# Transcriptome analysis of bread wheat leaves in response to salt stress

**Nazanin Amirbakhtiar[1,2], Ahmad Ismaili[1], Mohammad-Reza Ghaffari[3], Raheleh Mirdar Mansuri[3], Sepideh Sanjari[3], Zahra-Sadat Shobbar**[3]*

1 Plant Production and Genetic Engineering Department, Faculty of Agriculture, Lorestan University, Khorramabad, Iran, 2 National Plant Gene Bank of Iran, Seed and Plant Improvement Institute (SPII), Agricultural Research, Education and Extension Organization (AREEO), Karaj, Iran, 3 Department of Systems Biology, Agricultural Biotechnology Research Institute of Iran (ABRII), Agricultural Research, Education and Extension Organization (AREEO), Karaj, Iran

* shobbar@abrii.ac.ir

**Data Availability Statement:** Data Availability The raw transcriptome reads generated in the current study are available in the SRA (Sequence Read Achieve) of NCBI (Accession numbers of

## Abstract

Salinity is one of the main abiotic stresses limiting crop productivity. In the current study, the transcriptome of wheat leaves in an Iranian salt-tolerant cultivar (Arg) was investigated in response to salinity stress to identify salinity stress-responsive genes and mechanisms. More than 114 million reads were generated from leaf tissues by the Illumina HiSeq 2500 platform. An amount of 81.9% to 85.7% of reads could be mapped to the wheat reference genome for different samples. The data analysis led to the identification of 98819 genes, including 26700 novel transcripts. A total of 4290 differentially expressed genes (DEGs) were recognized, comprising 2346 up-regulated genes and 1944 down-regulated genes. Clustering of the DEGs utilizing Kyoto Encyclopedia of Genes and Genomes (KEGG) indicated that transcripts associated with phenylpropanoid biosynthesis, transporters, transcription factors, hormone signal transduction, glycosyltransferases, exosome, and MAPK signaling might be involved in salt tolerance. The expression patterns of nine DEGs were investigated by quantitative real-time PCR in Arg and Moghan3 as the salt-tolerant and susceptible cultivars, respectively. The obtained results were consistent with changes in transcript abundance found by RNA-sequencing in the tolerant cultivar. The results presented here could be utilized for salt tolerance enhancement in wheat through genetic engineering or molecular breeding.

## Introduction

Plant growth and productivity are seriously threatened by abiotic stresses [1]. Among abiotic stresses, salt stress is considered a serious threat to crop yield worldwide [2]. Wheat is the third most important cereal crop in the world [3], and salinity levels of 6–8 dsm-1 cause to decline wheat yield [4]. A practical approach to minimize salinity's impact on global wheat production is to enhance salt tolerance in wheat cultivars.

Ion toxicity, nutrient limitations, and oxidative and osmotic stresses are the adverse effects of salinity stress on crops [5]. Plant salt tolerance is achieved through integrated responses at

SRR7975953, SRR7968059, SRR7968053, and SRR7920873). All the rest of relevant data are within the manuscript and its Supporting information files.

**Funding:** Z-S.S. received the grant from Iran National Science Foundation (INSF Grant Number: 96000095) and Agricultural Biotechnology Research Institute of Iran (ABRII Grant Number: 24-05-05-010-960594). The funders had no role in study design, data collection and analysis, decision to publish, or preparation of the manuscript.

**Competing interests:** The authors have declared that no competing interests exist.

physiological, cellular, molecular, and metabolic levels. At the molecular level, genes coding for transcription factors, ion transporters, protein kinases, and osmolytes are involved in salt tolerance [6, 7]. Some signaling pathways, including plant hormones, salt overly sensitive (SOS), calcium, mitogen-activated protein kinase (MAPK), and proline metabolism, play critical roles in salt stress tolerance, as well [8–12]. Salinity tolerance, as a quantitative trait, is under the control of multiple genes [13]. Thus, it is necessary to discover key components underlying the salt tolerance network to improve it through genetic engineering.

RNA-sequencing provides a much more accurate measurement of transcript levels and isoforms compared to other transcriptomic methods [14]. A few studies applied RNA-sequencing technology to inspect the transcriptome profile of shoots under salt conditions in bread wheat in recent years. Comparing the shoot expression profiling in a salinity tolerant mutant of *Triticum aestivum* L and its susceptible wild type exposed to salt stress resulted in discovering some salt tolerance involved genes like polyamine oxidase, arginine decarboxylase, and hormones-associated genes, which were further up-regulated in the mutant. They also succeeded in finding "Butanoate metabolism" as a novel salt stress-response pathway and indicated that oxidation-reduction (redox) homeostasis was essential for salt tolerance [15]. In another study, Mahajan et al. (2017) performed RNA-sequencing to prepare transcriptome profiling of flag leaves in the salt-tolerant cultivar of Kharcha in response to salt stress. They indicated that the up-regulated genes under salt stress were related to different biological processes like ion transport, phytohormones signaling, signal transduction, osmoregulation, flavonoid biosynthesis, and ROS homeostasis [16]. Luo et al. (2019) compared young and old leaf transcriptome of a salt-tolerant bread wheat cultivar and a high-yielding cultivar with lower salt tolerance in response to salinity. They found that the polyunsaturated fatty acid (PUFA) metabolism was the most significant term/pathway in the salt-tolerant wheat cultivar according to the enriched GO terms and the Kyoto Encyclopedia of Genes and Genomes (KEGG) pathways analysis. They suggested that PUFAs could promote salt tolerance through the photosynthetic system and JA-related pathways [17].

Zhang et al. (2016) compared root transcriptome response of a salt-tolerant and a salt-sensitive cultivar and identified two NAC transcription factors (TFs), a MYB TF (homologous to *AtMYB33*), a gene positively associated with root hair development (*Ta.RSL4*) and a gene coding for histone-lysine N-methyl transferase (homologous to Arabidopsis *AtSDG16*) as essential genes for salinity tolerance in *Triticum aestivum* [18]. Amirbakhtiar et al. (2019) evaluated transcriptome profile of a salt tolerant bread wheat cultivar in response to salinity. They identified pathways related to transporters, phenylpropanoid biosynthesis, TFs, glycosyltransferases, glutathione metabolism and plant hormone signal transduction as the most important pathways involved in salt stress response [19]. Mahajan et al. (2020) sequenced root transcriptome of a salt tolerant wheat cultivar at anthesis stage. They showed that genes involved in ROS homeostasis, ion transport, signal transduction, ABA biosynthesis and osmoregulation up-regulated in response to salt stress. They also indicated that genes coding for expansin, dehydrins, xyloglucan endotransglucosylase and peroxidases, engaged in root growth improvement, up-regulated under salt stress [20]. Despite the valuable insight discovered by recent researches about the cellular and molecular mechanisms engaged in salinity stress response and tolerance in bread wheat, many aspects are still uncovered. In the current study, considering Iran as one of the origin lands of *Triticum aestivum* and its wild lineages [21–24], deep transcriptome sequencing was used for an Iranian salt-tolerant wheat cultivar (Arg) under normal and salinity conditions to complement the insights regarding molecular mechanisms involved in bread wheat salt-tolerance. We succeeded in providing a panel of the regulatory mechanisms at transcriptional level in the leaves of the salt-tolerant wheat cultivar (Arg) under salinity stress by

identifying all differentially expressed genes, novel salt-responsive genes, and diverse metabolic pathways involved in response to salinity stress.

## Materials and methods

### Wheat culture conditions and salinity treatment

Seeds of the bread wheat salt-tolerant (Arg) and salt-sensitive (Moghan3) genotypes were kindly supplied by Seed and Plant Improvement Institute (SPII), Karaj, Iran. After surface sterilizing the seeds in 1% sodium hypochlorite, they were grown on moist filter paper for approximately 72 hours. The uniform germinated seeds were then selected and transferred to half-strength Hoagland's culture solution in the greenhouse. NaCl solution (150 mM) was used to treat the three-week old plants for 12 and 72 hours. The leaves of the control and salt-stressed plants were collected separately. The number of biological replicates was four, and each replicate included three independent plants. The samples were frozen instantly in liquid nitrogen and kept at -80˚C.

### Measurements of $Na^+$ and $K^+$ concentrations

The leaves of the plants exposed to salt stress for 72 hr were harvested and dried at 70˚C for 48 hr. Flame spectrophotometry method was used to measure $Na^+$ and $K^+$ concentrations [25].

### RNA isolation and Illumina sequencing

RNA was extracted from wheat leaves with four biological replicates under normal and salinity conditions utilizing RNeasy Plant Mini Kit (Qiagen). Equal quantities of the total RNA of every two biological replicates of Arg cultivar were pooled together to prepare two replicates for the RNA sequencing. Agarose gel electrophoresis, nanodrop, and Agilent Bioanalyzer 2100 system (Agilent Technologies Co. Ltd., Beijing, China) were used to control the quantity, quality, and integrity of RNA. The RIN value of the samples used for sequencing was more than or equal to 6.9. cDNA library preparation and sequencing were performed using an Illumina Hiseq 2500 platform at the Novogene Bioinformatics Institute (Beijing, China). The generated reads were paired-end with 150bp size. After sequencing, adapter-containing reads, poly-N-containing reads (N > 10%), and low quality (Qscore< = 5) base-containing reads were eliminated.

### Read mapping and reference-based assembly

The FastQC toolkit was used to assess the quality of raw fastq data. Tophat software with standard parameters was utilized to map the high-quality reads to the wheat reference genome (ftp://ftp.ensemblgenomes.org/pub/release-34/plants/fasta/triticum_aestivum/dna/). Cufflinks with default settings was applied to create assembly based on the Tophat mapping files. The individual assemblies were merged using Cuffmerge with default options, and a final assembly was produced. The novel transcripts were recognized via Cuffmerge [26]. For functional annotation, Blast2GO via BlastX with $1e^{-3}$ as an e-value cut-off was used to align all the transcripts against NCBI's non-redundant protein database. This software was also utilized to obtain the gene ontology (GO) terms of transcripts with a p-value cut-off of 0.05.

### Differential gene expression analysis

The gene/transcript expression was calculated using the FPKM method. Significantly differentially expressed genes (DEGs) were identified applying Cuffdiff provided in the Cufflinks package based on $|\log_2$ fold change$| \geq 1$ and Q-value cut off $\leq 0.01$.

### Functional annotation and pathway enrichment analysis of DEGs

The Online KEGG Automatic Annotation Server (KAAS), http://www.genome.jp/kegg/kaas [27], was utilized to identify metabolic pathways in which DEGs were engaged. The pathway analysis of the DEGs was performed applying Mapman (version 3.5.1; http://mapman.gabipd.org/web/guest) [28] with a p-value limit of $\leq 0.05$.

### Confirmation of RNA-sequencing results by Real-Time PCR analysis

The extracted RNAs from three biological replicates were reverse-transcribed with qScript cDNA Synthesis Kit (Quantabio, USA) for first-strand cDNA synthesis based on the manufacturer's protocol. LightCycler® 96 Real-Time PCR System (Roche Life Science, Germany) and SYBR Premix EX TaqII (Takara Bio Inc, Japan) were utilized to perform Quantitative Real-Time PCR. Actin was used in place of an internal control gene in the RT-PCR experiment to normalize the gene expression value [17, 19, 29, 30]. Genes with $\log_2$ fold change $\geq 1$ or $\log_2$ fold change $\leq -1$ were considered as significant DEGs.

Specific primers for the selected genes are listed in S1 Table. Each gene's transcript level in every genotype under control conditions was utilized as the calibrator for each time point. The $2^{-\Delta\Delta Ct}$ procedure [31] was applied to calculate the relative expressions of the selected genes.

## Results

### Na⁺ and K⁺ content

The salt stress led to a significant increase in $Na^+$ content and a significant decrease in the $K^+/Na^+$ ratio in the roots and leaves of both genotypes. A significant increase was observed in the $K^+$ content of the roots in both genotypes under salinity stress, while no significant change was observed in their leaves. As expected, less $Na^+$ content and more $K^+/Na^+$ ratio were observed in the leaves of the tolerant cultivar (Arg) compared to those of the susceptible cultivar (Moghan 3) under salinity stress. A higher accumulation and maintenance of $Na^+$ ion in the roots than in the leaves may act as a tolerance mechanism by maintaining the essential osmotic potential for absorbing water into the root and limiting $Na^+$ ion flux into the shoot [32]. The higher accumulation of $Na^+$ ion in the root than in the leaf blade is a salt tolerance mechanism in the grasses, limiting the transfer of sodium ions into photosynthetic cells and active meristem tissues [33]. The higher amount of $Na^+$ in the roots of Arg compared to those of Moghan 3 and the less amount of $Na^+$ in the leaves of Arg compared to those of Moghan 3 under salinity stress indicate that root can be considered as an important barrier to prevent the transfer of $Na^+$ to the leaves in Arg (S1 and S2 Figs). This result is consistent with the results obtained by Davenport et al. (2007) [34].

### Sequencing statistics and reference-based analysis

A total of 114.29 million raw reads were obtained by transcriptome sequencing. After removing adapters and low-quality reads, a total of 112.6 million clean reads were produced, while more than 88.1% of clean reads had Phred-like quality scores at the Q30 level (Table 1). Accession numbers of SRR7975953, SRR7968059, SRR7968053, and SRR7920873 at the SRA (Sequence Read Achieve) of NCBI include the raw transcriptome reads generated in the current study.

Mapping the cleaned high-quality reads to the wheat reference genome (ftp://ftp.ensemblgenomes.org/pub/release-34/plants/fasta/triticum_aestivum/dna/) showed that around 81.9%-85.7% of the reads were mapped successfully to the wheat reference sequence,

**Table 1. Summary of sequencing results.**

| Sample name | Raw reads (paired end) | Clean reads (paired end) | Q20% | Q30% |
|---|---|---|---|---|
| Control-rep1 | 27,152,094 | 26,623,849 | 96.38 | 91.25 |
| Control-rep2 | 31,085,137 | 30,489,825 | 96.38 | 91.23 |
| Salt-stressed-rep1 | 26,752,355 | 26,460,382 | 94.69 | 88.1 |
| Salt-stressed-rep2 | 29,307,102 | 29,030,761 | 94.37 | 87.5 |
| Total | 114,296,688 | 112,604,817 | $\geq$ 94.37 | $\geq$ 87.5 |

including 72.8%-79.3% uniquely matched (Table 2). The aligned reads were assembled using cufflinks while 187003 and 98819 transcript isoforms and genes were identified, respectively.

## Exploration of novel transcripts via mRNA sequencing

The discovery of novel genes/transcripts is one of the main benefits of RNA- sequencing experiments [14, 35, 36]. The current study identified 27800 and 16339 novel transcript isoforms and genes, respectively. Conforming with other crops, including rice and maize [37, 38], the mean length of the novel transcripts (1609 bp) was less than that of the annotated transcripts (2304 bp). Based on the gene ontology analysis results, a putative function was assigned to more than 53.1% of the novel transcripts.

The GO analysis of the novel transcripts revealed that these novel genes would play a role in biological processes, including stimulus responses, localization, biogenesis, and biological regulation (S2 Table). Molecular function classification showed that the novel transcripts were enriched in some terms such as transferase, oxidoreductase, catalytic, and hydrolase activities; small molecule and ion binding; carbohydrate derivative binding; and organic cyclic and heterocyclic compound binding (S3 Table). The novel transcripts were also enriched in some cellular component categories, such as intracellular membrane-bounded organelle, an integral component of membrane, cytoplasm, mitochondrion, nucleus, and chloroplast (S4 Table).

## Identification of DEGs involved in salt stress response

In total, 4290 genes were differentially regulated under salinity stress, of which 2346 and 1944 were up- and down-regulated genes, respectively (S5 Table). Among the DEGs, 110 and 98 genes were exclusively expressed under salt-stressed and normal conditions, respectively (S3A Fig). Some essential genes engaged in response to abiotic stresses, including LEA proteins, dehydrins, bHLH transcription factor, phosphatase 2C, peroxidase, and calcium-transporting ATPase plasma membrane-type (S6 Table), were observed among the genes exclusively expressed under salinity stress. Surveying the fold change distribution of the DEGs indicated

**Table 2. Results of mapping reads to the reference genes.**

| Reads mapping | Reads number (%) | | | |
|---|---|---|---|---|
| | Control-rep1 | Control-rep2 | Salt-stressed-rep1 | Salt-stressed-rep2 |
| Total reads | 53247698 | 60979650 | 52920764 | 58061522 |
| Total mapped reads | 45573294(85.6%) | 52244911(85.7%) | 43386060(82%) | 47556787(81.9%) |
| Unique match | 42239930(79.3%) | 48007353(78.7%) | 40090198(75.8%) | 42283991(72.8%) |
| Multi-position match | 3333364(6.3%) | 4237558(7%) | 3295862(6.2%) | 5272796(9.1%) |
| Total unmapped reads | 7674404 (14.4%) | 8734739(14.3%) | 9534704(18%) | 10504735(18.1%) |

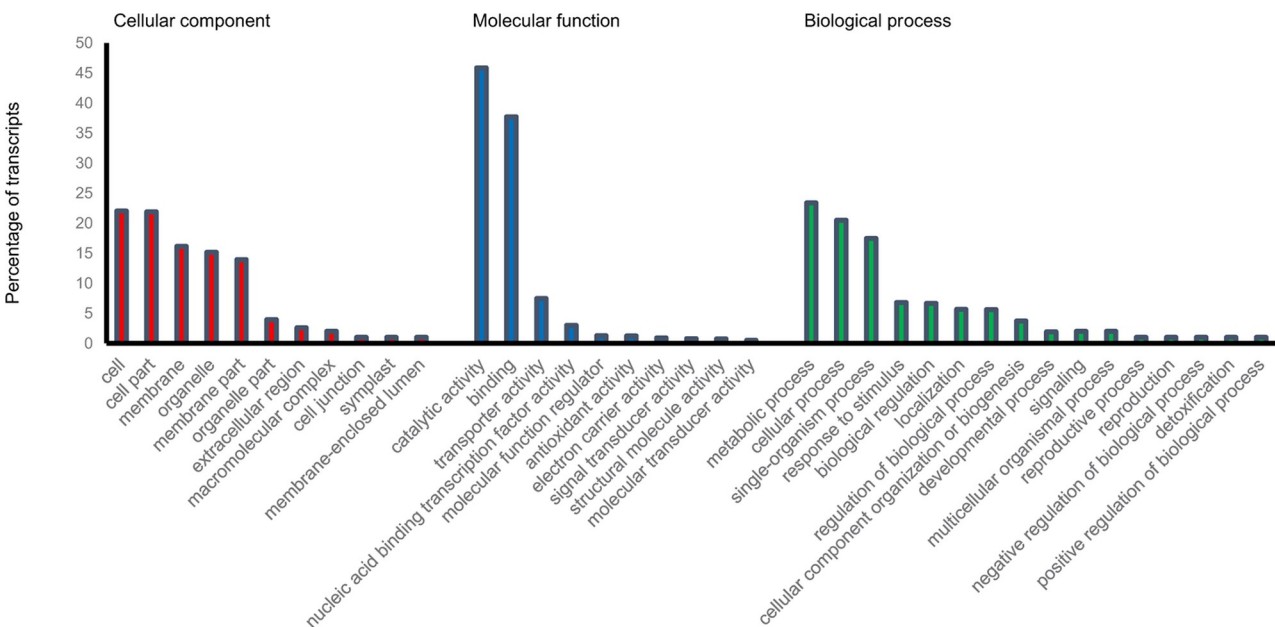

**Fig 1. GO classification of the DEGs in Arg cultivar.** Percentage of the transcripts in cellular component, molecular function, and biological process classifications are represented.

that the most and least number of the genes had a fold change of 2–3 and 6–7, respectively (S3B Fig).

## GO categorization of DEGs

The gene ontology analysis indicated that GO terms were assigned to 3594 out of 4290 genes. In biological process classification, the majority of the genes were involved in the metabolic process (23.4%) followed by cellular process (20.4%), single-organism process (17.4%), stimulus-response (6.7%), and biological regulation (6.6%). In molecular function categorization, catalytic activity (45.9%), binding (37.7%) and transporter activity (7.5%) were the most frequent terms and in the cellular component category, cell (22%), cell part (21.9%), and membrane (16.1%) were the most dominant terms (Fig 1).

## Functional identification of novel DEGs

Comparing the functional annotation of the novel salt responsive genes against NCBI's nonredundant (nr) protein database utilizing the Blast2GO tool indicated that 320 (70%) out of the 457 novel DEGs were aligned to the NR protein database. In contrast, the rest of the genes (30%) displayed no homology to database sequences. The GO classification showed that 230 novel DEGs (50.3%) were assigned to GO terms, and 205 novel DEGs (44.9%) were grouped in significant GO terms (S4A Fig). In the cellular component category, cell part, cell, and membrane were the prevailing clusters. However, the top three classes were catalytic activity, binding, and transporter activity concerning the molecular function. In biological process categorization, most of the genes were involved in the metabolic and cellular processes followed by regulating the single-organism process, regulating the biological process and responding to a stimulus (S4B Fig).

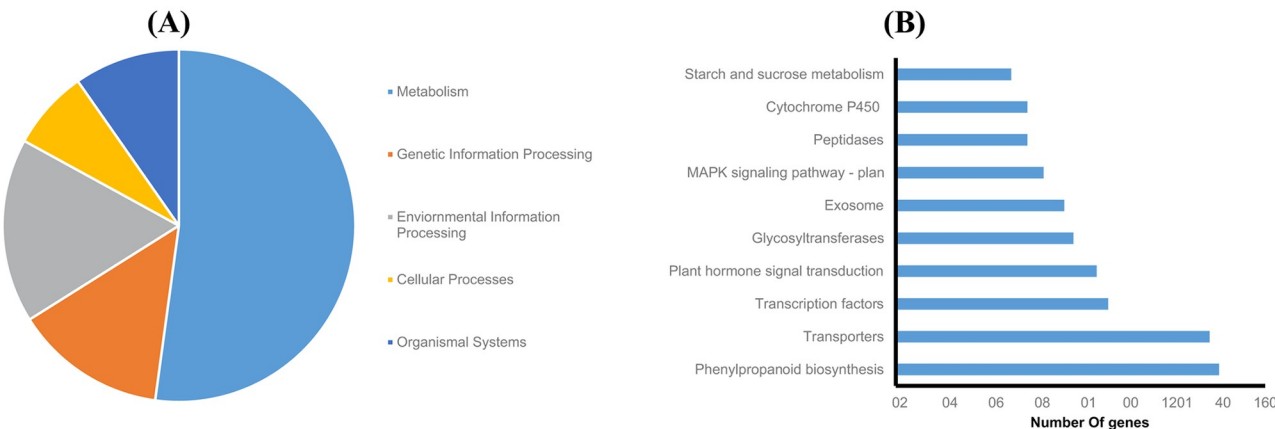

**Fig 2. KEGG categorization of the DEGs.** (A) Categorization of the DEGs into five chief KEGG classes. (B) The top 10 pathways with the highest gene number.

The genes coding for Phosphatase 2C [39], Delta-1-pyrroline-5-carboxylate synthase [40], MYB transcription factor [41], Sodium/Calcium exchanger [42], Ribulose bisphosphate carboxylase large chain [43], Late Embryogenesis Abundant protein [44], Glutathione S-trasferase [45], and cytochrome P450 monooxygenase [46] (S7 Table) with potential roles in salt stress response were observed among the novel DEGs.

## KEGG pathway classification of DEGs

In an attempt to map DEGs to various biological pathways, a single-directional BLAST search showed that 1503 out of the 4290 DEGs were categorized into 227 KEGG pathways, located in the five chief KEGG classes (Fig 2A). Pathways relating to phenylpropanoid biosynthesis, transporters, transcription factors, plant hormone signal transduction, glycosyltransferases, exosome, MAPK signaling pathway, peptidases, cytochrome P450, and sucrose and starch metabolism included the highest number of DEGs (Fig 2B, S8 Table). The involvement of these pathways in environmental stress response was confirmed in previous reports [29, 47, 48].

The phenylpropanoid pathway with the highest gene number is responsible for synthesizing diverse secondary metabolites in plants such as lignin, flavonoids, and coumarins playing roles in developmental and stress–associated processes [49, 50]. In the first step of this pathway, cinnamic acid is synthesized from phenylalanine by the rate-limiting enzyme of phenylalanine ammonia-lyase (PAL) [51]. In this study, 29 up-regulated DEGs coding for PAL were mapped in this pathway.

Plants utilize deposition of lignin or modification of monomeric lignin composition in the cell wall to defeat salinity stress [52]. In the present study, the up-regulated DEGs coding for shikimate hydroxycinnamoyl transferase, cinnamoyl-CoA reductase, and caffeic acid 3-O-methyltransferase, which were all involved in lignification, were mapped in the phenylpropanoid pathway, while their over-expression was also reported under salinity stress in prior researches [53–55].

## Functional analysis of salt-regulated genes using Mapman

The putative function of the salt-regulated genes was searched utilizing Mapman to visualize salt-induced alterations in diverse metabolic processes. Metabolic pathway overviews based on

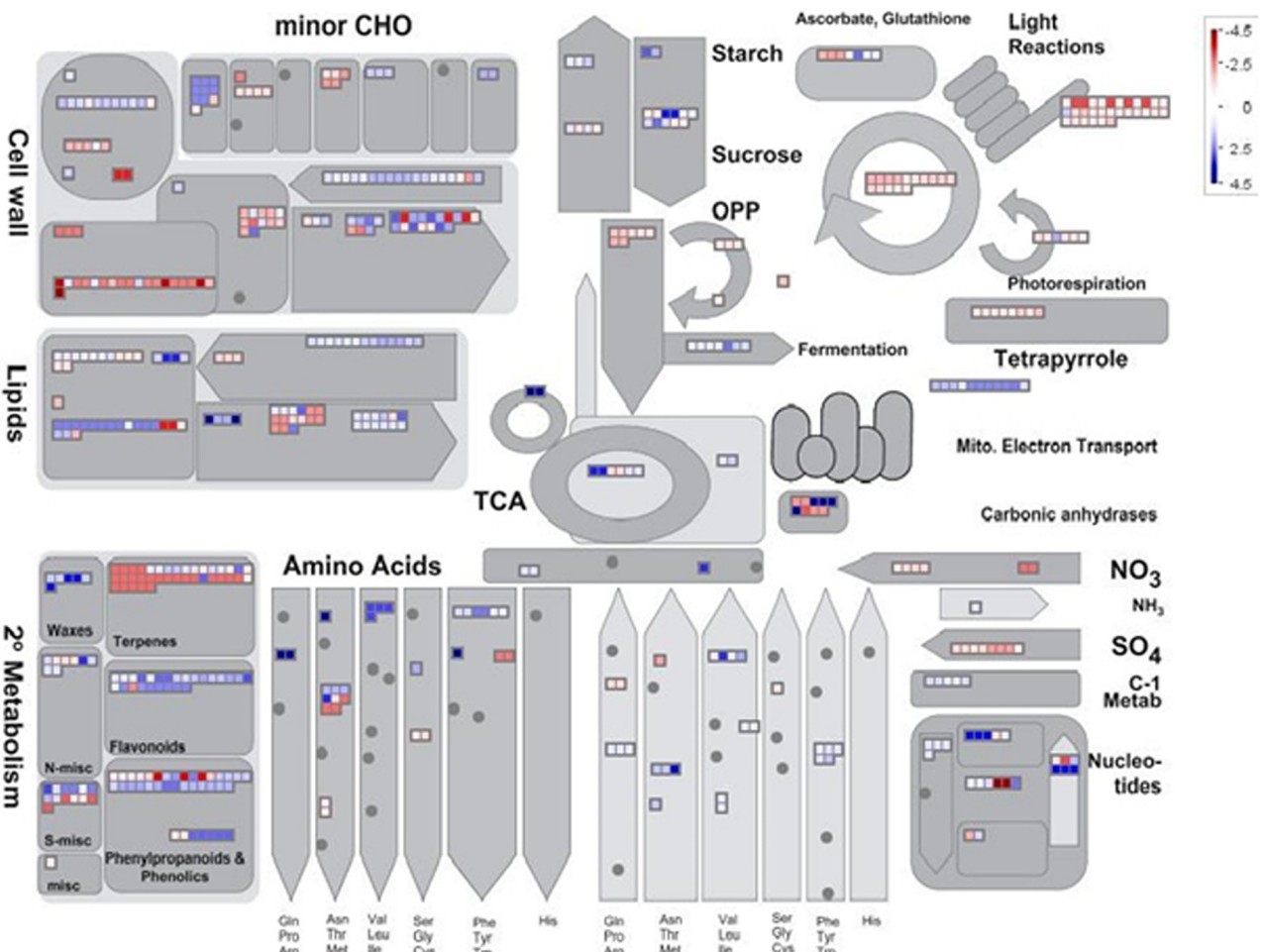

**Fig 3. Metabolic pathway overview of the DEGs in Arg cultivar under salinity stress utilizing Mapman.** The down- and up-regulated genes are shown in red and blue, respectively.

the results of mapping salt-responsive genes indicated that photosynthesis and cell wall biosynthesis pathways were among the enriched pathways (Fig 3, S9 Table). Most of the genes encoded chlorophyll-binding proteins in the photosynthesis pathway, showing down-regulation under salt stress. The decrease in photosynthesis efficiency under abiotic stresses was reported in previous studies [56, 57].

Mapping the DEGs to the cellular pathways indicated that the misc pathway, including genes regarding abiotic stress-related various enzyme families, was enriched under salt stress (S5 Fig., S9 Table). Most of the misc pathway genes are Germin-like proteins (GLPs), which code for ubiquitous plant glycoproteins and belong to the cupin superfamily. One of the main roles of the proteins mentioned above is triggering the abiotic stress-tolerance in many plant species. Li et al. (2016) revealed that GLP transcripts were plentiful after treatment with high salinity, PEG6000, abscisic acid, and methyl viologen in soybean. Arabidopsis plants overexpressing a GLP from soybean indicated enhanced drought, salt, and oxidative tolerance [58]. Furthermore, Arabidopsis transgenic plants, which overexpressed genes encoding peanut GLPs, showed increased tolerance to salinity. Complementary studies also showed that PR-

defense genes and antioxidant coding genes, which can increase salt tolerance, showed up-regulation in transgenic plants [59].

Investigating the secondary metabolite pathways revealed that the genes playing roles in terpenoid, lignin, phenols, isoflavonoid, and wax metabolic pathways were significantly enriched under salt stress (S6 Fig, S9 Table). Furthermore, the stress response pathways showed that the transcription regulators and peroxidases and the genes relating to brassinosteroid signaling pathways were enriched in Arg cultivar under salt stress (S7 Fig., S9 Table).

## Confirmation of gene expression patterns by qRT-PCR

The expression pattern of nine candidate salt-regulated genes was examined by qRT-PCR to validate the RNA-sequencing results (Fig 4). The high consistency between qRT-PCR and RNA sequencing results was observed ($R^2$ = 0.98), confirming the identified DEGs in the present research. The candidate genes' expression profile was assessed in the two salt contrasting genotypes to obtain further insight. Based on the obtained results, *Ta.bHLH35*, *Ta.CIPK23*, and *Ta.P5CS* were up-regulated significantly in the tolerant cultivar after 12 hr of salt stress, while the increase in the expression of these genes was much less in the sensitive cultivar than in the tolerant cultivar and was not significant (Fig 4A, 4B and 4F). *Ta.ERF061* showed significant up-regulation after 12 hr of exposure to salt stress in both cultivars. However, at the time

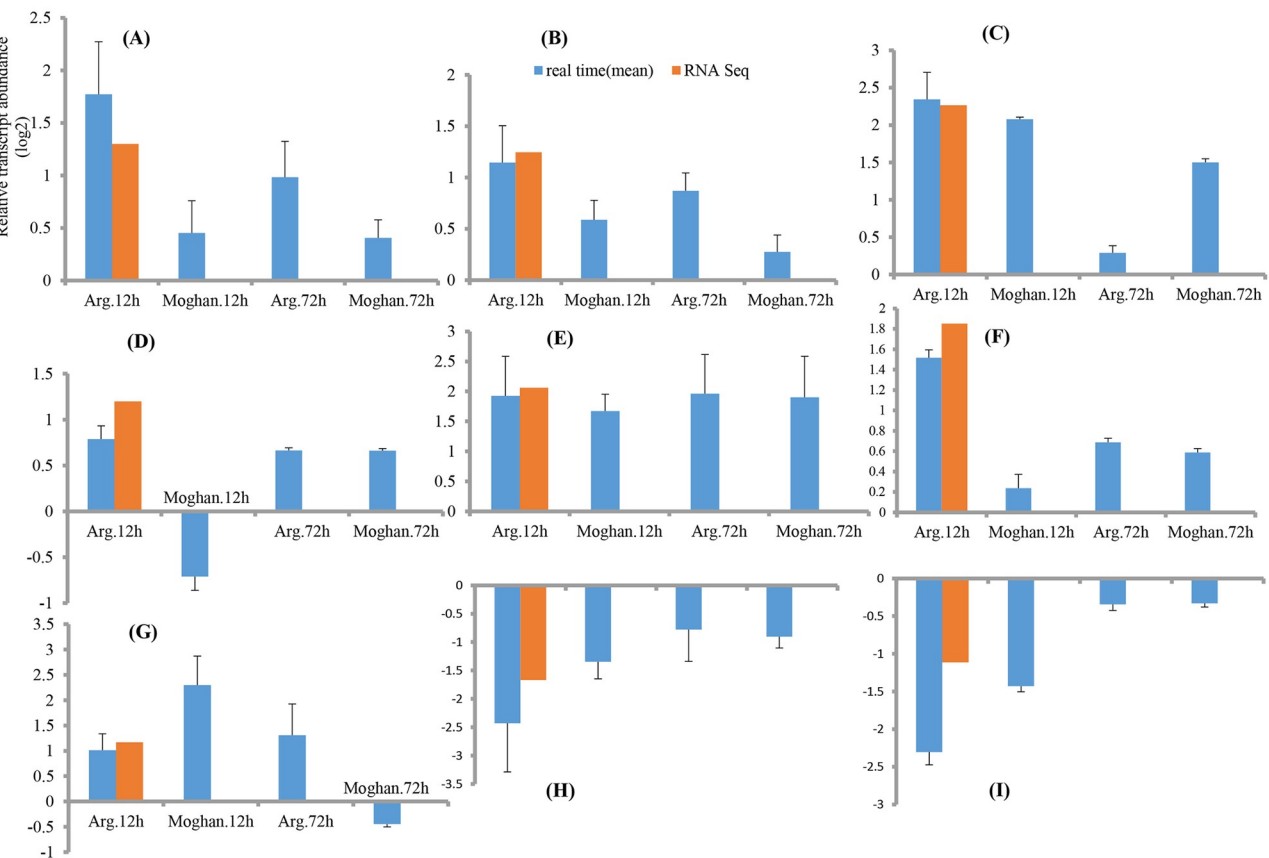

**Fig 4.** Validation of the candidate genes by qRT-PCR including bHLH transcription factor 35 (A), calcineurin B-like protein (CBL)-interacting protein kinase 23 (B), ethylene responsive factor 061 (C), heat shock transcription factor B1 (D), NAC transcription factor (E), pyrroline-5-carboxylate synthetase (F), salt response protein (G), Ribuluse biphosphate carboxylase small chain (H), and Phosphoglycerate Kinase (I). Refer to S1 Table to find the gene ensemble IDs.

point of 72 hr, a more severe decrease in expression was observed in the tolerant cultivar compared to the sensitive cultivar, which can be related to the quicker response of the tolerant cultivar to salt stress. (Fig 4C). For *Ta.HSFB1* at the time point of 12 hr, while the tolerant cultivar indicated up-regulation, the sensitive cultivar showed down-regulation (Fig 4D). For another transcription factor, *Ta.NAC*, significant up-regulation was observed in both cultivars at the two-time points, and there was no significant difference between the cultivars (Fig 4E). For the gene encoding salt response protein, while the tolerant cultivar showed up-regulation at the two-time points, the sensitive cultivar indicated up-regulation after 12 hr of exposure to salt treatment and down-regulation after 72 hr of exposure to salinity (Fig 4G). Furthermore, for the gene coding for RUBISCO small chain involved in photosynthesis, a more severe decrease was observed in the tolerant cultivar compared to the sensitive cultivar (Fig 4H). The decrease in this gene expression may be due to the need to change the energy flow from the biosynthesis of photosynthesis-engaged macromolecules toward respiratory paths to supply the energy needed to overcome stress. A regulatory gene called phosphoglycerate kinase, with a possible negative effect on stress tolerance, showed more severe down-regulation in the tolerant cultivar than in the sensitive cultivar after 12 hr of exposure to salt stress (Fig 4I).

## Discussion

Next-generation sequencing technologies with the ability to characterize transcriptome profiles of different organisms under different conditions can reveal the molecular basis of salt stress response in plants. In general, genes engaged in salt stress response can be divided into three classes, comprising stress sensing and signaling-related genes, transcriptional regulators, and salinity-stress associated genes [60].

Signal transduction paths play crucial roles in the response of plants to different stresses. Variation in cytosol's calcium concentration is one of the early responses to various stimuli, and calcium transporting elements actively maintain this flux and homeostasis [61]. In the current research, two genes encoding calcium-transporting ATPases were up-regulated under salt stress. One of them is a novel gene (represented as *Ta.ACA7* in Fig 5 and S10 Table), expressed only under salt stress. Orthologous of the forenamed gene in rice, *Os. ACA7* (Os10g0418100), is activated by calmodulin (CaM) [61]. $Ca^{2+}$-ATPases are involved in maintaining $Ca^{2+}$ homeostasis [61], and the up-regulation of them has been observed in different plant species, including tomato, tobacco, Arabidopsis, and soybean, in response to salinity stress [62–65]. Genes coding for glutamate receptors (GLRs), known as non-selective cation channels that can be engaged in $Ca^{+2}$ transport [61], were also observed among the DEGs in this study (Fig 5, S10 Table). Glutamate receptors are responsive to abiotic stresses based on the previous reports [66, 67]. After an increase in the $Ca^{2+}$ concentration under salinity, CBL-interacting protein kinases (CIPKs) [68] with the ability to transduce the signal to downstream protein activity and gene transcription may become activated [69]. Among the DEGs, 13 genes coding for CIPKs were discovered. One gene coding for CaM was up-regulated in response to salinity (Fig 5, S10 Table). CaM, known as a $Ca^{2+}$-sensing protein, is involved in the transduction of $Ca^{2+}$ signals. Conformational changes occur in CaM after interacting with $Ca^{2+}$, and then, CaM influences the activities of the proteins which bind to it. Several CaM-binding proteins are engaged in plant responses to salinity stress, showing that CaM plays a central role in stress adaptation in plants [70]. A differentially expressed CaM-binding gene in the current study is *Ta.MLO* (Fig 5, S10 Table), encoding a plant-specific seven-transmembrane domain protein. A study on the MLO family in rice concluded that environmental stresses might provoke alteration in the H2O2 level via interaction between MLO and CaM. The resulting H2O2 might act as a

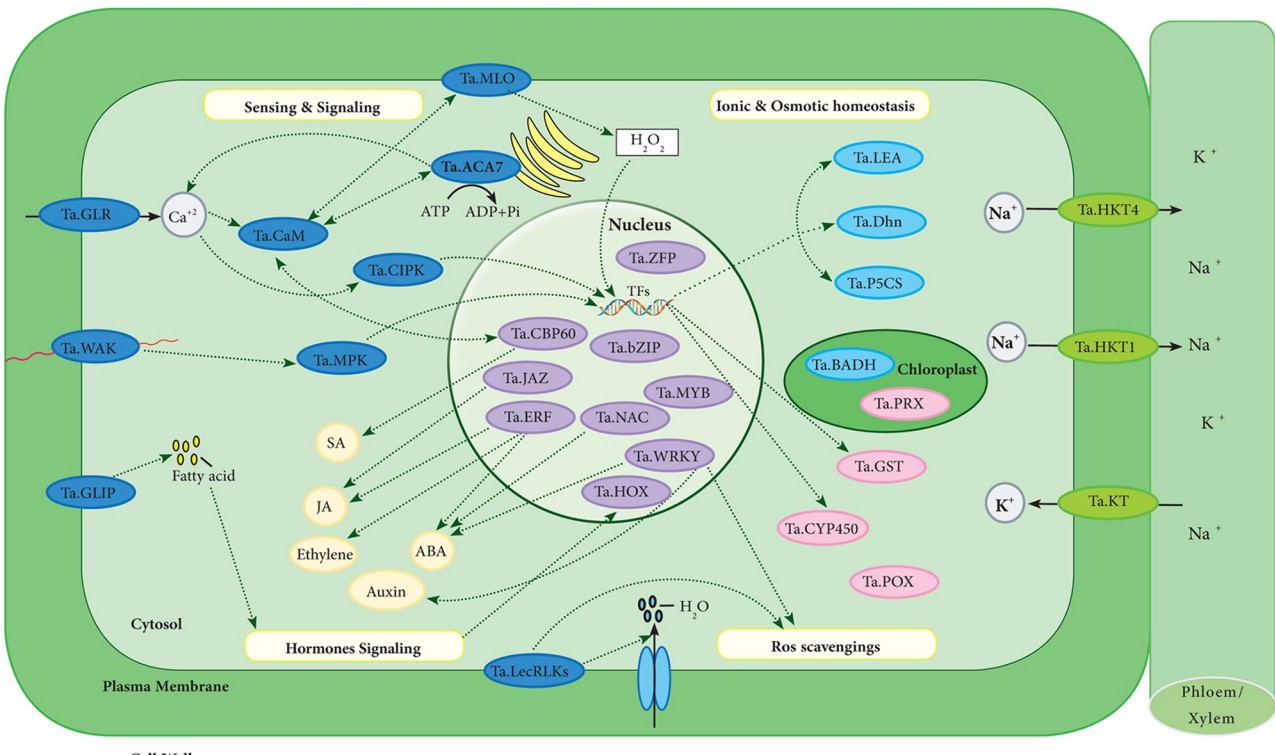

**Fig 5. The model proposed for a response to salinity stress in Arg cultivar.** Dark blue and purple colors were used to exhibit stress sensing and signaling-related genes and transcription factors, respectively. Light blue, light green, and pink colors were utilized to depict genes involved in the reaction to osmotic, ionic, and oxidative stresses caused by salinity, respectively.

messenger to stimulate the responsive genes' expression to acclimatize to the stress [71]. *CBP60*, a CaM-binding transcription factor, was up-regulated under salinity stress in the present study (Fig 5, S10 Table). It has been shown that the overexpression of *CBP60* (At5g26920) in Arabidopsis resulted in increased defense response, hypersensitivity to ABA, and drought tolerance, possibly through activating salicylic acid accumulation [72].

Previous reports indicated that the overexpression of GDSL esterase lipases (GLIPs) could release fatty acids acting as hormone signal transduction molecules [73]. It has also been reported that excessive GLIPs exhibited improved salinity stress tolerance in *Oryza sativa* and *Arabidopsis thaliana* [74, 75]. Five genes coding for Ta.GLIPs were up-regulated under salinity stress in the present study (Fig 5, S10 Table).

Receptor-like kinases (RLKs), as the largest gene family in plants, play crucial roles in signaling networks [76]. Wall-associated kinases (WAKs), as a subfamily of RLKs, function as a signaling linker between the cytoplasm and the extracellular region [77]. It has been reported that WAKs are engaged in regulating plant adaptation to abiotic stresses. Arabidopsis plants overexpressing *AtWAK1* showed increased aluminum tolerance [78], and Arabidopsis plants with the impaired expression of *AtWAKL4* indicated more hypersensitivity to excessive Na+, K+, Cu+2, and Zn+2 [79]. In the current study, six genes coding for WAKs were up-regulated under salt stress (Fig 5, S10 Table). LecRLKs, another subfamily of RLKs, can be engaged in salinity tolerance, including a plasma membrane-localized LecRLK from *Pisum sativum*. Tobacco plants overexpressing *PsLecRLK* showed enhanced salt tolerance by increasing ROS

scavenging activity and activating water channels, leading to reduced ROS accumulation and enhanced water uptake [80]. In the present research, three genes coding for LecRLKs were up-regulated in response to salinity stress (Fig 5, S10 Table).

Many TFs were observed among the DEGs, indicating their crucial roles in salt stress response. They regulate the expression of downstream genes liable for salinity stress tolerance in plants. ERFs, bZIPs, Zn-fingers, NACs, MYBs, and WRKYs were found among the differentially expressed TFs, and some of them were discussed here.

MYB TFs are known as one of the largest and most diverse families of TFs in plants [81, 82]. The involvement of MYB TFs in salt tolerance has been reported in previous studies [83, 84]. Twenty-seven genes coding for MYBs were observed among the DEGs in the present research (Fig 5, S10 Table).

Plant basic leucine zipper (bZIP) TFs are involved in regulating abiotic stress signaling pathways mediated by abscisic acid (ABA) in plants [85]. Tomato *SlbZIP38* regulates drought and salinity tolerance negatively via regulating ABA signaling [86]. The overexpression of cotton *GhABF2*, encoding a bZIP TF, significantly increased tolerance to drought and salinity in Arabidopsis and cotton [87]. Two genes coding for bZIPs were differentially expressed in the current study (Fig 5, S10 Table).

Four families of zinc finger proteins (ZFP), including C2H2, CCCH, C3HC4, and C4, have crucial roles in regulating phytohormone and stress response in plants [88]. The engagement of zinc finger TFs in salt tolerance has been reported in previous studies. Transgenic rice overexpressing OsZFP213 indicated improved salt tolerance via enhancing ROS scavenging ability [89]. Tobacco plants overexpressing GhZFP1, a CCCH-type zinc finger protein from cotton, showed increased tolerance to salinity stress and resistance to Rhizoctonia solani [90]. In the present study, around 17 differentially expressed zinc finger TFs were identified (Fig 5, S10 Table).

TIFY proteins are engaged in regulating many plant processes, including response to stresses. JAZ proteins, working as the jasmonic acid signaling pathway's key regulators, are the best-characterized sub-group of TIFY proteins. Two genes coding for TIFY were found among the DEGs (Fig 5, S10 Table). The involvement of TIFY TFs in wheat salt tolerance was reported in a previous study [91].

In the present study, 31 genes coding for WRKY TFs were differentially expressed under salt stress, among which only one gene showed down-regulation (S10 Table). WRKY TFs are engaged in increasing salinity tolerance in plants via regulating stomatal conductance, ROS levels, and auxin and ABA signaling [92].

In addition, 28 NAC domain-containing genes were differentially regulated under salt stress in the current study, among which only four genes were down-regulated (Fig 5, S10 Table). NAC TFs take part in complicated signaling networks related to stress response in plants [93]. Rice *OsNAC022*, induced by drought, high salinity, and ABA, enhanced drought and salinity stress tolerance via regulating an ABA-dependent pathway in transgenic plants [94]. *TsNAC1* from a halophyte called *Thellungiella halophila* targeted positive ion transportation regulators and improved salt tolerance in both *T. halophila* and Arabidopsis [95].

Some ethylene response factors (ERFs) bind to dehydration-responsive elements, function as a central regulatory hub, and incorporate ethylene, abscisic acid, jasmonate, and redox signaling in abiotic stress response in plants [96]. In the present study, 15 genes relating to ERF transcription factors were differentially expressed under salinity stress (S10 Table). Previous studies have shown that the overexpression of ERFs by increasing salt-responsive genes' expression leads to salt tolerance in plants [97, 98].

We also identified transcripts encoding homeodomain-containing transcription factors (HOX) 7 and 22, which were significantly up-regulated under salt stress (Fig 5, S10 Table).

According to the previous reports, the HOX family members as regulators of plant growth and development were remarkably enriched in NaCl-induced transcripts in *Oryza sativa* [99, 100]. It has also been reported that ABA, GA, SA, and auxin enhance the transcript levels of some HOXs [99].

A high ratio of cytosolic $K^+/Na^+$ is necessary to keep ionic homeostasis under stress and increases salinity tolerance in wheat (Oyiga et al., 2016). Plants utilize various methods at different levels to retain this ratio in the cytosol. One selected approach in plants is sending out $Na^+$ from the roots. SOS1, a plasma membrane $Na^+/H^+$ antiporter, drives $Na^+$ out from the root. Evaluating the transcriptome response of the root in Arg cultivar under salt stress showed the up-regulation of *SOS1* under salinity stress [19]. Sustaining a high ratio of $K^+/Na^+$ in the cell cytoplasm can also be performed by sequestrating $Na^+$ into the vacuoles of root cells done by the tonoplast $Na^+/H^+$ antiporter (NHX1). Hexokinase1 phosphorylates NHX1 and increases its stability [101]. Transcriptome response analysis of the root in Arg cultivar under salt stress showed a significant increase in the expression of hexokinase1 [19]. Evaluating the transcriptome response of leaves to salt stress showed that 22 genes involved in transporting sodium, potassium, or both significantly responded to salt stress. Among the up-regulated sodium transporters in leaves, a gene indicated severe up-regulation under salt stress (represented as *Ta.HKT1* in Fig 5 and S10 Table). An orthologue of the mentioned gene in Arabidopsis, At4g10310, encodes the sodium transporter HKT1. This transporter shows a central role in plant tolerance to salinity. It loads $Na^+$ into the phloem sap in shoots and unloads it in roots, leading to eliminating large quantities of $Na^+$ from the shoot [102]. The other sodium transporter is represented as *Ta.HKT4* in Fig 5 and S10 Table. The orthologue of the forenamed gene in rice is Os04g0607500 that encodes for the cation transporter HKT4. *OsHKT4* acts as a low-affinity sodium transporter and is possibly engaged in regulating $K^+/Na^+$ homeostasis [103]. Seven genes, coding for potassium transporters, were differentially expressed under salinity stress. Among the potassium transporters, we can refer to a gene represented as *Ta.KT* in Fig 5 and S10 Table. The orthologue in Arabidopsis, At2g30070, encodes AtKT1 and acts as a high-affinity potassium transporter [104].

In order to deal with the osmotic stress caused by salinity, the genes encoding for LEA proteins (*Ta.LEA*) and dehydrins (*Ta.Dhn*) as well as the genes involved in biosynthesis of organic osmolytes like proline (pyrroline-5-carboxylate synthase; *Ta.P5CS*) and glycine betaine (betaine aldehyde dehydrogenase; *Ta.BADH*) were up-regulated under salinity stress (Fig 5, S10 Table).

Furthermore, peroxiredoxin (*Ta.PRX*), peroxidases (*Ta.POX*), Cytochrome P450 (*Ta. CYP450*), and Glutathione-S- transferases (*Ta.GST*) were differentially regulated in reaction to oxidative stress arising from salinity (Fig 5, S10 Table). Cytochrome P450 (CYP) includes a superfamily of heme-containing proteins that take part in redox homeostasis and numerous biosynthetic pathways [105]. Eight genes belonging to the CYP71 family, as the largest CYP family in plants, were up-regulated in the present research. This result is in line with those obtained by another study on transcriptome response of wheat leaf to salinity stress [16], indicating the CYP71 family might play a role in salinity stress tolerance.

## Supporting information

**S1 Fig. Comparison of means for changes in Na+ content (a), K+ content (b) and K+/Na + ratio (c) in the leaves of Arg and Moghan3 under the normal and salt stressed conditions at the probability level of 5%.**
(DOCX)

**S2 Fig. Comparison of means for changes in Na+ content (a), K+ content (b) and K+/Na + ratio (c) in the roots of Arg and Moghan3 under the normal and salinity treated conditions at the probability level of 5%.**
(DOCX)

**S3 Fig. Survey of DEGs observed between the control and salt treated samples (cut off p-value: 0.01).** (a) Out of 4290 DEGs, 110 and 98 genes were exclusively expressed under the salt stress (STL) and control (CL) conditions, respectively (b) Fold change distribution of 4082 DEGs present in both normal and salt treated samples.
(DOCX)

**S4 Fig.** (a) Annotation statistics of the novel DEGs. (b) GO classification of the novel DEGs under salt stress.
(DOCX)

**S5 Fig. Cellular pathway overview of DEGs in *T. aestivum* under salinity stress using Mapman.** Blue: up-regulated genes and red: down-regulated genes.
(DOCX)

**S6 Fig. Secondary metabolite pathway overview of the DEGs in *T. aestivum* under salinity stress using Mapman.** Blue: up-regulated genes and red: down-regulated genes.
(DOCX)

**S7 Fig. Stress response pathways overview of the DEGs in *T. aestivum* under salinity stress using Mapman.** Blue: up-regulated genes and red: down-regulated genes.
(DOCX)

**S1 Table. The primers used for Real Time PCR.**
(XLSX)

**S2 Table. Biological process classification of the novel transcripts.**
(XLSX)

**S3 Table. Molecular function classification of the novel transcripts.**
(XLSX)

**S4 Table. Cellular component classification of the novel transcrips.**
(XLSX)

**S5 Table. List of the differentially expressed genes.**
(XLSX)

**S6 Table. List of the genes exclusively expressed under salt stress.**
(XLSX)

**S7 Table. List of the novel differentially expressed genes.**
(XLSX)

**S8 Table. KEGG pathway classification of the DEGs.**
(XLSX)

**S9 Table. Results of functional analysis of the salt-regulated genes using Mapman.**
(XLSX)

**S10 Table. The genes applied in the model.**
(XLSX)

## Acknowledgments

The authors are grateful to Seed and Plant Improvement Institute (SPII) for providing the seeds, Miss. Saeedeh Asari for her technical assistance and Mr. Mohammad Jedari to help in creating the artworks.

## Author Contributions

**Conceptualization:** Zahra-Sadat Shobbar.

**Data curation:** Nazanin Amirbakhtiar.

**Formal analysis:** Nazanin Amirbakhtiar, Mohammad-Reza Ghaffari, Raheleh Mirdar Mansuri.

**Funding acquisition:** Zahra-Sadat Shobbar.

**Investigation:** Nazanin Amirbakhtiar.

**Methodology:** Nazanin Amirbakhtiar, Zahra-Sadat Shobbar.

**Project administration:** Zahra-Sadat Shobbar.

**Supervision:** Ahmad Ismaili, Zahra-Sadat Shobbar.

**Validation:** Nazanin Amirbakhtiar, Zahra-Sadat Shobbar.

**Visualization:** Nazanin Amirbakhtiar, Raheleh Mirdar Mansuri.

**Writing – original draft:** Nazanin Amirbakhtiar.

**Writing – review & editing:** Ahmad Ismaili, Sepideh Sanjari, Zahra-Sadat Shobbar.

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
