## [Decision Letter · Decision Letter 0]

22 Apr 2021

PONE-D-21-06731

Transcriptome analysis of bread wheat leaves in response to salt stress

PLOS ONE

Dear Dr. Shobbar,

Thank you for submitting your manuscript to PLOS ONE. After careful consideration, we feel that it has merit but does not fully meet PLOS ONE’s publication criteria as it currently stands. Therefore, we invite you to submit a revised version of the manuscript that addresses all the points raised by reviewers.

We look forward to receiving your revised manuscript.

Kind regards,

Guangxiao Yang, Ph.D

Academic Editor

PLOS ONE

Journal Requirements:

2. Please include a copy of Table 1 which you refer to in your text on page 7.

Please include a copy of Table 2 which you refer to in your text on page 8.

Please include a copy of Table 11 which you refer to in your text on page 19.

Reviewers' comments:

Reviewer's Responses to Questions

**Comments to the Author**

1. Is the manuscript technically sound, and do the data support the conclusions?

Reviewer #1: Yes

Reviewer #2: Yes

2. Has the statistical analysis been performed appropriately and rigorously? 

Reviewer #1: Yes

Reviewer #2: Yes

3. Have the authors made all data underlying the findings in their manuscript fully available?

Reviewer #1: Yes

Reviewer #2: Yes

4. Is the manuscript presented in an intelligible fashion and written in standard English?

Reviewer #1: Yes

Reviewer #2: Yes

5. Review Comments to the Author

Reviewer #1: Authors studied two varieties of wheat plant under salt stress using physiological and molecular (deep sequencing and qRT-PCR) aspects. The studied varieties are extreme considering their tolerance to salinity. They found several differentially expressed genes in salt tolerance variety. Enriched biological pathways of stress responsive genes are studied in the current study. Finally, authors confirmed deep sequencing outputs using real-time PCR. There are some comments that authors should consider:

1. Why deep transcriptome sequencing not studied for Moghan variety while evaluated physiologically and also by real-time PCR.

2. Please describe in the material and methods that which variety is utilized for deep sequencing.

3. It is not described in the material and methods that how Na and K content are measured.

4. Please clarify in the in the material and methods that how you found significant DEGs in real-time PCR.

5. It is not clear why Actin is used as an internal control gene. Dose it previously used for salt stress in wheat or other plants? Reference is required.

6. TopHat software description should be explained in material and methods instead of results (page 8 )

7. Are you sure that mentioned Gene Ensembl Id (column 2 of Table S1) are belonged to Ensembl? I could not found them at Ensembl.

8. It is not clarified in the S2, S3 and S4 tables that which GOs are enriched in salinity or normal conditions.

9. Gene Accession Id is necessary for S2, S3 and S4 tables. Please add this column.

10. Column 1 of Table S10 is not clear. If sequences are aligned after blast use “aligned” term instead of “Blasted”.

11. Some values of log(Fc) of table S10 are obscure. Please check rows 25, 47, 93 and … . likely some values of description column are not clear ([Hordeum vulgare vulgare], ---NA---).

12. Ven diagram is not appropriate choice for showing DEGs in Fig S3(a). if you are insisted on showing your data using Ven diagram then remove “4082”. But it is better to use another type of graphs.

13. Fig S3(b) is ambiguous for common values. Use > or < signs along with values.

14. Please use the correct terms instead of “Blasted” and “No Blast” in Fig S4A. Additionally, terms of manuscript (page 11, paragraph 1) are not coincided with FigS4A.

Reviewer #2: COMMENTS FOR THE AUTHOR:

Transcriptome analysis of bread wheat leaves in response to salt stress

The manuscript is well written and presents new insights regarding the wheat leaves transcriptome exposed to salinity. Minor corrections are suggested.

In introduction, wheat transcriptome analysis of other plant tissues in salinity besides leaves can also be cited to make it comprehensive. Abbreviations, when used for the first time in text, should be given in brackets with the complete forms. A few suggestions are given as under.

Page 2, line 16: The word “Soil” may be removed. It is better to begin the sentence with salinity.

Page 3, line 37: The first paragraph of “Introduction” needs improvement due to the repetition of words like yield and salinity.

Page 9, line 196-199: Revise the statement.

In figures, please use an appropriate font size to make the text more visible especially the X and Y axis in curves are not clear.

6. PLOS authors have the option to publish the peer review history of their article (what does this mean?). If published, this will include your full peer review and any attached files.

Reviewer #1: **Yes: **Behnam Bakhshi

Reviewer #2: **Yes: **Fariha Khan

---

## [Author Response · Author response to Decision Letter 0]

1 Jun 2021

Dear Dr. Guangxiao Yang, 

Respected Academic Editor of PLOS ONE,

Thanks to you and the respected reviewers for the time and fruitful comments. We revised the manuscript “PONE-D-21-06731” and carefully addressed issues raised by the reviewers, as you may find in the following and in the revised manuscript with track change.

I hope the manuscript would be accepted for publication in this revised version.

Best regards

Zahra-Sadat Shobbar

Agricultural Biotechnology Research Institute of Iran (ABRII) 

Seed and Plant Improvement Institutes Campus 

P. O. Box: 31535-1897 

Mahdasht Road, Karaj, Iran 

Phone: +98 26 32703536 

Fax: +98 26 32704539

1. Why deep transcriptome sequencing not studied for Moghan variety while evaluated physiologically and also by real-time PCR.

Response 1. Actually, we were interested to do deep transcriptome sequencing for both varieties, but unfortunately our country (Iran) is under international sanctions and our research budget is very limited, so we didn’t succeed to do transcriptome sequencing for the sensitive genotype, however in order to be able to compare the salinity response of the two genotypes to some extent, physiological analysis and Real Time PCR for some important genes were done.

2. Please describe in the material and methods that which variety is utilized for deep sequencing.

Response 2. It was described in the material and methods that which variety was utilized for deep sequencing.

3. It is not described in the material and methods that how Na and K content are measured.

Response 3. The method used for measuring Na+ and K+ content was described in the materials and methods.

4. Please clarify in the material and methods that how you found significant DEGs in real-time PCR.

Response 4. Genes with log2 fold change ≥ 1 or log2 fold change ≤-1 were considered as significant DEGs and it was mentioned in the material and methods.

5. It is not clear why Actin is used as an internal control gene. Dose it previously used for salt stress in wheat or other plants? Reference is required.

Response 5. Yes, Actin has been used as an internal control gene in other studies conducted on gene expression in wheat under salt stress [1-4]. Some references were added to the manuscript. 

6. TopHat software description should be explained in material and methods instead of results (page 8 )

Response 6. Description about TopHat software was deleted from the results. It was already explained in the material and methods.

7. Are you sure that mentioned Gene Ensembl Id (column 2 of Table S1) are belonged to Ensembl? I could not found them at Ensembl.

Response 7. We used the previous reference genome of Triticum aestivum, TGACv1, which was the reference genome available in ensembl plants at the time we analyzed our data, but now it is available in Ensembl plants archive and it is accessible to it using the following address http://oct2017-plants.ensembl.org/Triticum_aestivum/Info/Index

8. It is not clarified in the S2, S3 and S4 tables that which GOs are enriched in salinity or normal conditions.

Response 8. The mentioned (S2, S3 and S4) tables present the Gene ontology of all the novel transcripts (regardless of their expression pattern). In fact, in this study we succeeded to find 187003 transcripts (related to 98819 genes) in the assembly, among which 27800 transcripts (related to 16339 genes) were novel. As there were no annotation for the novel genes, we were going to annotate these novel genes through GO analysis. We used GO analysis to find what biological processes, molecular functions and cellular components the discovered novel genes are involved in. These novel transcripts might be expressed under normal or stressed conditions or both of them. Identification of all novel transcripts available in an assembly and assigning GO terms to them regardless of their expression profile has been conducted in other studies [1, 5, 6].

9. Gene Accession Id is necessary for S2, S3 and S4 tables. Please add this column.

Response 9. Given that the transcripts presented in these tables are novel transcripts, they have no accession Id, but their position on the chromosome is known and the corresponding position was presented for each transcript.

10. Column 1 of Table S10 is not clear. If sequences are aligned after blast use “aligned” term instead of “Blasted”.

Response 10. The term "Blasted" was given by the blast2go software, however, the term "Blasted" was changed to "aligned" in column 1, based on the respectful reviewer’s opinion. Additionally, we should apologize for the mistake in numbering table S10, so we changed its number to S5 to be in accordance with the text.

11. Some values of log(Fc) of table S10 are obscure. Please check rows 25, 47, 93 and … . likely some values of description column are not clear ([Hordeum vulgare vulgare], ---NA---).

Response 11. Log2 FC is calculated by cuffdiff package; when the expression is observed exclusively under the stressed condition and expression value under normal condition is 0, Log2 FC is displayed by inf and conversely, when the expression is observed only under normal condition and expression value under stressed condition is 0, Log2 FC is displayed by #NAME?. Description about inf and #NAME? was added at the bottom of the table. In addition, unclear descriptions in the description column were corrected but" ---NA---" in this column means that the related sequence doesn’t code a protein and no result has been obtained for that sequence using blastx. In addition, as it was mentioned in the response 10, we should make an apology for the mistake in numbering table S10, so we changed its number to S5 to be in accordance with the text.

12. Ven diagram is not appropriate choice for showing DEGs in Fig S3(a). if you are insisted on showing your data using Ven diagram then remove “4082”. But it is better to use another type of graphs.

Response 12. In Fig S3(a), we are going to show how many DEGs are expressed in common under both normal and salt stressed conditions (4082 ) and how many DEGs are exclusively expressed under normal (98) or salt stressed (110) conditions. We would appreciate if the respected reviewer let us know which type of diagram might be appropriate choice for this aim rather than Venn diagram. Using Venn diagrams for such a purpose has been observed in other studies [1, 2]. 

13. Fig S3(b) is ambiguous for common values. Use > or < signs along with values.

Response 13. Necessary changes were made.

14. Please use the correct terms instead of “Blasted” and “No Blast” in Fig S4A. Additionally, terms of manuscript (page 11, paragraph 1) are not coincided with FigS4A.

Response 14. In fact, the terms "Blasted" and "No Blast" are the terms given by the software used for blasting against non redundant protein database of NCBI, Blast2go. However, based on the respected reviewer’s suggestion and in order to apply a more suitable term having more accordance with the text, the terms "Blasted" and "No Blast" were replaced by "Aligned" and "No Homology" terms in FigS4A.

15. In introduction, wheat transcriptome analysis of other plant tissues in salinity besides leaves can also be cited to make it comprehensive. 

Response 15. Based on the respected reviewer’s suggestion, studies conducted on transcriptome analysis of other tissues of wheat were also added.

16. Abbreviations, when used for the first time in text, should be given in brackets with the complete forms

Response 16. Necessary changes were made.

17. Page 2, line 16: The word “Soil” may be removed. It is better to begin the sentence with salinity.

Response 17. Yes, the respected reviewer is right. It is done.

18. Page 3, line 37: The first paragraph of “Introduction” needs improvement due to the repetition of words like yield and salinity. 

Response 18. Necessary changes were made.

19. Page 9, line 196-199: Revise the statement.

Response 19. The statement was revised.

20. In figures, please use an appropriate font size to make the text more visible especially the X and Y axis in curves are not clear. 

Response 20. Necessary changes were made.

21. Please include a copy of Table 11, which you refer to in your text on page 19.

Response 21. Please excuse us for a mistake in nomination of "Table 11". In fact, "Table 11" must be "Table S10", so we corrected "Table 11"to "Table S10" in the manuscript.

References

1. Amirbakhtiar N, Ismaili A, Ghaffari MR, Nazarian Firouzabadi F, Shobbar Z-S. Transcriptome response of roots to salt stress in a salinity-tolerant bread wheat cultivar. PloS one. 2019;14(3):e0213305.

2. Goyal E, Amit SK, Singh RS, Mahato AK, Chand S, Kanika K. Transcriptome profiling of the salt-stress response in Triticum aestivum cv. Kharchia Local. Scientific reports. 2016;6(1):1-14.

3. Luo Q, Teng W, Fang S, Li H, Li B, Chu J, et al. Transcriptome analysis of salt-stress response in three seedling tissues of common wheat. The Crop Journal. 2019;7(3):378-92.

4. Ma X, Gu P, Liang W, Zhang Y, Jin X, Wang S, et al. Analysis on the transcriptome information of two different wheat mutants and identification of salt-induced differential genes. Biochemical and biophysical research communications. 2016;473(4):1197-204.

5. Mansuri RM, Shobbar Z-S, Jelodar NB, Ghaffari MR, Nematzadeh G-A, Asari S. Dissecting molecular mechanisms underlying salt tolerance in rice: a comparative transcriptional profiling of the contrasting genotypes. Rice. 2019;12(1):1-13.

6. Shankar R, Bhattacharjee A, Jain M. Transcriptome analysis in different rice cultivars provides novel insights into desiccation and salinity stress responses. Scientific reports. 2016;6(1):1-15.

---

## [Decision Letter · Decision Letter 1]

22 Jun 2021

Transcriptome analysis of bread wheat leaves in response to salt stress

PONE-D-21-06731R1

Dear Dr. Shobbar,

We’re pleased to inform you that your manuscript has been judged scientifically suitable for publication and will be formally accepted for publication once it meets all outstanding technical requirements.

Kind regards,

Guangxiao Yang, Ph.D

Academic Editor

PLOS ONE

Additional Editor Comments (optional):

Reviewers' comments:

Reviewer's Responses to Questions

**Comments to the Author**

1. If the authors have adequately addressed your comments raised in a previous round of review and you feel that this manuscript is now acceptable for publication, you may indicate that here to bypass the “Comments to the Author” section, enter your conflict of interest statement in the “Confidential to Editor” section, and submit your "Accept" recommendation.

Reviewer #1: All comments have been addressed

Reviewer #2: All comments have been addressed

2. Is the manuscript technically sound, and do the data support the conclusions?

Reviewer #1: Yes

Reviewer #2: Yes

3. Has the statistical analysis been performed appropriately and rigorously? 

Reviewer #1: Yes

Reviewer #2: Yes

4. Have the authors made all data underlying the findings in their manuscript fully available?

Reviewer #1: (No Response)

Reviewer #2: Yes

5. Is the manuscript presented in an intelligible fashion and written in standard English?

Reviewer #1: Yes

Reviewer #2: Yes

6. Review Comments to the Author

Reviewer #1: (No Response)

Reviewer #2: Authors have revised the manuscript carefully by addressing comments and suggestions. The revised manuscript is acceptable.

7. PLOS authors have the option to publish the peer review history of their article (what does this mean?). If published, this will include your full peer review and any attached files.

Reviewer #1: **Yes: **Behnam Bakhshi

Reviewer #2: **Yes: **Fariha Khan

---

## [Editor Report · Acceptance letter]

28 Jun 2021

PONE-D-21-06731R1 

Transcriptome analysis of bread wheat leaves in response to salt stress 

Dear Dr. Shobbar:

I'm pleased to inform you that your manuscript has been deemed suitable for publication in PLOS ONE. Congratulations! Your manuscript is now with our production department. 

Kind regards, 

on behalf of

Dr Guangxiao Yang 

Academic Editor

PLOS ONE